# Overexpression of a *Malus baccata* (L.) Borkh WRKY Factor Gene *MbWRKY33* Increased High Salinity Stress Tolerance in *Arabidopsis thaliana*

**DOI:** 10.3390/ijms26125833

**Published:** 2025-06-18

**Authors:** Xinhui Wang, Ming Gao, Yihan Kong, Qian Yu, Lu Yao, Xingguo Li, Wenhui Li, Wanda Liu, Ruining Hou, Lihua Zhang, Deguo Han

**Affiliations:** 1Key Laboratory of Biology and Genetic Improvement of Horticultural Crops (Northeast Region), Ministry of Agriculture and Rural Affairs / National-Local Joint Engineering Research Center for Development and Utilization of Small Fruits in Cold Regions, College of Horticulture & Landscape Architecture, Northeast Agricultural University, Harbin 150030, China; wxh18846915926@163.com (X.W.); 13029799003@163.com (M.G.); kyihan0731@126.com (Y.K.); 20050726@163.com (Q.Y.); 15114709942@163.com (L.Y.); xingguoli@neau.edu.cn (X.L.); wenhuili@neau.edu.cn (W.L.); 2Horticulture Branch, Heilongjiang Academy of Agricultural Sciences, Harbin 150069, China

**Keywords:** *Malus baccata*, *MbWRKY33*, high-salinity stress, genetic transformation, transcriptional regulation

## Abstract

The WRKY transcription factor family is a significant family of plant transcription factors (TFs). Plant growth and development are often influenced by abiotic factors, such as salinity and low temperature. Numerous studies have demonstrated that WRKY TFs primarily influence plant responses to adversity. However, there are few studies on the role of WRKY genes in the stress responses of *Malus baccata* (L.) Borkh. We cloned the *MbWRKY33* gene from *Malus baccata* for this research, and its roles in salt stress tolerance were analyzed. Phylogenetic tree analysis revealed that MbWRKY33 and PbWRKY33 have the highest homology. Subcellular localization revealed that MbWRKY33 was located within the nucleus. An analysis of tissue-specific expression showed that *MbWRKY33* had relatively high expression levels in young leaves and roots. Moreover, *Arabidopsis thaliana* plants overexpressing *MbWRKY33* exhibited stronger resistance to salt stress compared with the wild type (WT) and the unloaded line empty vector (UL). Under the treatment of 200 mM NaCl, transgenic *Arabidopsis thaliana* plants exhibited significantly higher activities of antioxidant enzymes like superoxide dismutase (SOD), peroxidase (POD), and catalase (CAT) than the control. In contrast, the WT and the UL lines had elevated levels of malondialdehyde (MDA) and reactive oxygen species (ROS). In addition, *MbWRKY33* elevates transgenic plant resistance to salt stress by regulating the expression levels of *AtNHX1, AtSOS1*, *AtSOS3, AtNCED3, AtSnRK2,* and *AtRD29a*. Results indicated that *MbWRKY33* in *Malus* might be linked to high-salinity stress responses, laying a foundation for understanding WRKY TFs’ reaction to such stress.

## 1. Introduction

Plants encounter diverse stressors that impact their growth, including biotic pressures like pathogen invasions and insect feeding, as well as abiotic environmental challenges, such as high salinity, drought, and low temperature [1]. Under diverse abiotic stresses, plants induce the expression of multiple genes, triggering a cascade of physiological and biochemical changes that enable active stress responses. High salinity and low temperature are prevalent abiotic environmental factors [2,3]. In agricultural settings, these stresses severely compromise the yield and quality of horticultural crops, even threatening national food security [4,5]. Specifically, high salinity stress impairs water uptake and causes ion toxicity, primarily due to the high osmotic potential generated by sodium (Na^+^) and chloride (Cl^−^) ions [6]. Low-temperature stress, conversely, damages root functions, disrupts stomatal closure and induces leaf wilting. Both stresses disrupt intracellular free radical metabolism, prompting the production of ROS. This leads to increased plant cell membrane permeability, electrolyte imbalance, and membrane lipid peroxidation—manifested by elevated levels of MDA and ROS [7]. When confronting biotic or abiotic stresses, plants first detect and interpret stress signals through a complex transduction network, initiating adaptive responses [8]. Finally, the transcription of stress-responsive genes occurs, inducing changes in physiological processes and facilitating stress signal transduction. These genes include signal-cascade-related genes, regulatory genes, and functional genes (which directly encode proteins involved in stress adaptation, such as enzymes for osmolyte synthesis or antioxidant defense), and various other genes associated with related pathways. Multiple pathways mediate plants’ responses to salt stress, and they involve the salt overly sensitive (SOS) pathway, the calcium-dependent protein kinase (CDPK) cascade reaction, the phospholipid pathway, the mitogen-activated protein kinase (MAPK) cascade reaction, and the abscisic acid (ABA) signaling pathway [9]. When plants respond to abiotic stresses, the network signal cascade reaction is realized using numerous TFs, and the WRKY transcription factor is among them.

TFs can modulate the expression of genes associated with stress responses, thus playing a crucial role in dealing with diverse abiotic stressors [10,11]. It has been found that members from transcription factor families like AP2/ERF, NAC, MYB, BZIP, and WRKY play key regulatory roles in abiotic stress responses. It has been verified that WRKY TFs play a role in responding to biotic and abiotic stresses as well as in the development processes of plants, including plant morphogenesis [12], seed germination [13], plant senescence, responses to external stresses [14], and hormonal signal transduction, etc. [15]. Compared with biotic stresses, there is relatively little information about the role of WRKY proteins in plants’ general abiotic stress responses. However, a large number of recent studies have indicated that WRKY transcription factor genes are involved in the responses of plants to abiotic stresses and ABA signals [16] and regulate the responses of plants to freezing, oxidative stress, drought, saline conditions, and low-temperature stress [17,18]. For instance, in *Arabidopsis thaliana*, *AtWRKY25* and *AtWRKY33* have been shown to regulate cold and salt tolerance by modulating the expression of stress-responsive genes [19,20]. *OsWRKY45* confers drought resistance through the ABA signaling pathway [21] while in tomatoes (*Solanum lycopersicum*). These studies highlight the conserved yet species-specific functions of WRKYs in abiotic stress adaptation. 

The WRKY TF family is named after the WRKY domain, which is approximately 60 amino acids long. It contains a highly conserved WRKYGQK amino acid sequence at its N-terminus and has a zinc finger-like motif at its C-terminus [22]. WRKY proteins usually contain one or two domains, and these domains show a high binding affinity for the TTGACC/T W-box element, which exists in the promoters of many defense-related genes and regulates their transcription. Thus, WRKY proteins control the expression of target genes by binding specifically to promoter regions with the W-box sequence (T)TTGAC(C/T) [23]. WRKY genes have received extensive attention over the past decade. For example, findings from the microarray analysis of WRKY genes in *Arabidopsis thaliana* indicate that they are highly responsive and regulated under salinity, drought, and cold stress conditions [24]. Cold stimulation in grapes causes at least 15 WRKY genes to display expression patterns induced by stress [25]. In addition, earlier studies have demonstrated that WRKY TFs in banana fruits play a role in ABA-mediated cold tolerance. This is achieved, at least partially, through the direct activation of NEC expression to boost ABA concentrations [26]. In *Arabidopsis thaliana*, heat and salt tolerance are regulated by WRKY25 and WRKY33 [27,28]. *AtWRKY25*, a WRKY gene in *Arabidopsis thaliana*, positively modulates the plant’s reaction to high-salinity and high-temperature stresses [29,30]. *CsWRKY2* (from *Camellia sinensi*) is involved in the ABA-signaling pathway and significantly influences plant responses to low-temperature and high-salinity stresses [31,32]. The *MdWRKY100* gene is the target of action for the transcription factor *MdSPL13*, thereby improving the salt endurance of apple plants [33]. However, the response mechanism of WRKY TFs to complex abiotic stress processes still needs further investigation.

*Malus baccata* is one of the tetraploid wild species that can be hybridized with cultivated apple types. It has excellent environmental adaptability and cold stress resistance [34]. Its extensive distribution across China coincides with regions known for apple cultivation. Therefore, this is a critical germplasm resource in apple and apple rootstock breeding. In addition to the effects of cold stress, the geographical spread and production of apples are mainly affected by abiotic stress environmental factors, such as drought and high salinity. So, breeding materials for low temperature and drought resistance is crucial for apple production. WRKY transcription factors are abundant in *Malus baccata*, and studies on WRKY transcription factors in *Malus baccata* have shown that they are involved in developmental processes, such as plant morphogenesis, seed germination, plant senescence, responses to external stresses, and hormonal signal transduction. Previous studies have reported that *AtWRKY33* is a paradigmatic salt-responsive transcription factor, with extensive molecular characterization underscoring its central role in abiotic stress adaptation. Under saline conditions, *AtWRKY33* orchestrates a multifaceted response by binding to W-box elements in the promoters of genes encoding key ion transporters, such as *SOS1* and *NHX1*, thereby enhancing cellular Na^+^ efflux and vacuolar sequestration [28,32]. The orthologous gene *OsWRKY33* in rice exhibits functional conservation yet species-specific adaptations in salt stress responses. In transgenic rice lines overexpressing *OsWRKY33*, enhanced salt tolerance is attributed to the coordinated upregulation of genes involved in osmolyte biosynthesis. Although previous studies have reported the involvement of WRKY33 in salt tolerance across multiple species, the regulatory mechanism of *MbWRKY33* in salt stress responses remains poorly understood in *Malus baccata*. In this study, using *Malus baccata* as the material, the *MbWRKY33* gene was cloned, and bioinformatics analysis was conducted on it. We employed real-time quantitative PCR (RT-qPCR) to examine the relative expression levels of *MbWRKY33* in various tissues of *Malus baccata*. Therefore, in-depth exploration and revelation of the response mechanism of the *MbWRKY33* transcription factor in *Malus baccata* in response to salt stress not only provides a theoretical basis and gene resources for the molecular breeding of salt-tolerant *Malus baccata* but also offers new research progress for grasping the association between plant stress and the processes of growth and development.

## 2. Results

### 2.1. Cloning and Bioinformatics Analysis of the MbWRKY33 Gene

The WRKY transcription factor *MbWRKY33* was obtained from *Malus baccata*, and Figure 1 illustrates the sequencing result. *MbWRKY33* has a full length of 1539 bp. An analysis by ExPASy-ProtParam showed that 512 amino acids make up the MbWRKY33 protein (Appendix A), among which Ser (S) (15.2%), Asn (N) (7.6%), Pro (P) (7.6%), and Gly (G) (6.2%) account for the highest proportions. With a molecular weight (MW) of 56775.04 kDa and a theoretical isoelectric point (pI) of 7.09, MbWRKY33 is a hydrophilic protein with a -0.967 grand average of hydropathicity (GRAVY), and this protein is unstable, with an instability coefficient of 54.65.

According to the sequence analysis, the MbWRKY33 protein possesses two conserved WRKY domains and falls within the class I WRKY family (Figure 1A). In this study, in comparing the amino acid sequence of the MbWRKY33 protein and the sequences of WRKY proteins from other species, a phylogenetic tree was constructed to explore the genetic relationship of MbWRKY33. It was determined that MbWRKY33 has a high similarity with PbWRKY33 (*Pyrus bretschneideri,* XP_048431322.1), MdWRKY33 (*Malus domestica,* NP_001281056.1), MsWRKY33 (*Malus sylvestris,* XP_050143309.1), and PaWRKY33 (*Prunus avium,* XP_021826437.1) (Figure 1B). 

The secondary structure of the protein encoded by the gene was predicted using SOPMA. The findings indicated that the secondary structure of MbWRKY33 consisted of 4.49% α-helix, 87.89% random coil, and 7.62% extended strand (Figure 2A). MbWRKY33 possesses two WRKY domains at amino acid positions 213–271 and 378–437, respectively, thus classifying it as a member of the WRKY family (Figure 2B). Additionally, the SWISS-MODEL website was utilized to predict the tertiary structure of the MbWRKY33 protein, and its structure corresponds to the forecasted secondary structure (Figure 2C).

### 2.2. MbWRKY33 Protein Was Located in the Nucleus

To investigate the subcellular localization of the *MbWRKY33* gene as a transcription factor, a fusion vector *MbWRKY33*-pCAMBIA1300 for transient expression in tobacco was constructed, with 35S::GFP used as a control. Under a fluorescence confocal microscope, the observation showed that the green fluorescent protein (GFP) in the control group was present located in both the plasmalemma and the nuclear compartment. At the same time, 35S::*MbWRKY33*::GFP was only localized in the nucleus. As shown in the figure (Figure 3). In addition, through observing the red fluorescence emanating from the nucleus, it can be confirmed that MbWRKY33 is a nuclear-localized protein.

### 2.3. Analysis of the Expression Level of MbWRKY33 in Malus baccata

To detect the tissue-specific expression of *MbWRKY33* in different parts of *Malus baccata* seedlings, we employed RT-qPCR for analyzing the expression of *MbWRKY33* in new leaves, stems, roots, and mature leaves. *MbWRKY33* exhibited the highest expression levels in new leaves and roots, whereas its expression levels in mature leaves and stems were comparably lower (Figure 4A). Under diverse stress conditions, such as 200 mM NaCl, 4 °C, 6% PEG6000, and 100 µM ABA, the expression of *MbWRKY33* was induced. During the initial 12-hour treatment period, the expression of *MbWRKY33* in mature leaves and roots peaked at comparable levels across different time points. As shown in Figure 4B,C, *MbWRKY33* was more sensitive to high salinity, low temperature, and drought stresses. Under the four stress conditions of high salinity, low temperature, drought, and ABA stress, the expression level of *MbWRKY33* in young leaves rose initially and subsequently declined. In young leaves, the peak expression times of *MbWRKY33* under various abiotic stress treatments were 3 h, 5 h, 7 h, and 3 h, respectively. In root tissues, *MbWRKY33* expression displayed a similar trend regardless of the specific stress applied, and the expression levels reached their peaks at 7 h, 3 h, 5 h, and 5 h, respectively. Following a 3-hour treatment with a high concentration of NaCl, the expression peaks of *MbWRKY33* in new leaves and roots were elevated to 13.5 times and 6.35 times the levels of the control group, respectively (Figure 4B,C). Compared with low temperature, drought, and ABA stresses, salt induced the expression of *MbWRKY33* more strongly, indicating that *MbWRKY33* is extremely sensitive to high salinity stress.

### 2.4. Overexpression of MbWRKY33 Enhances the Tolerance of Arabidopsis thaliana to Salt Stress

To reveal the role of *MbWRKY33* in the salt tolerance of *Arabidopsis thaliana*, we carried out vector resistance screening for the overexpressed *MbWRKY33* in the T1 generation transgenic *Arabidopsis thaliana* and the empty vector line (UL). Subsequently, RT-qPCR was used to measure the expression level of *MbWRKY33* in the T2 generation of the overexpressed transgenic *Arabidopsis thaliana*, with WT and UL plants as controls. DNA was extracted from leaf collection and verified via the PCR using specific (*MbWRKY33*-F/R, Appendix A); the result showed that all transgenic lines had targeted bands without WT and UL. Among them, the S1, S3, and S4 lines showed relatively high expression levels of *MbWRKY33* (Figure 5A). These three lines were carefully cultivated until the T3 generation, and subsequent abiotic stress treatments were carried out on the WT, UL, and transgenic plants (S1, S3, S4). Without any stress treatment, the growth of each line was essentially identical (Figure 5B). The WT, UL, S1, S3, and S4 plants were irrigated with a 200 mM NaCl solution for seven days, followed by the replacement of the irrigation solution with clean water. Phenotypic examination revealed that all plants exhibited some degree of wilting. The leaves of the seedlings of the transgenic lines (S1, S3, S4) remained green, with survival rates of 81%, 78%, and 77%, respectively. On the contrary, the survival rates of the WT and UL plants stood at 20% and 18%, respectively. Their leaves were severely affected, withering and turning yellow (Figure 5B). This is in sharp contrast to the equal survival rates of each line under the control conditions (Figure 5C). The experiment results suggest that, compared to the WT and UL, the overexpression lines had enhanced survival rates when exposed to high-salt conditions.

Under the control and high-salt treatment conditions, multiple important physiological parameters related to stress resistance, namely chlorophyll, proline, SOD, POD, and CAT, were detected at the same time. The results showed that no apparent differences in these indices among the overexpression lines, the WT lines, and the UL lines before the stress treatment. However, the overexpression lines were significantly different from the WT and UL varieties under the high-salt treatment. Higher amounts of chlorophyll and proline, as well as enhanced activities of SOD, POD, and CAT, were observed in them. In contrast, the contents of MDA, H_2_O_2_, O_2_^−^, and the relative electrical conductivity were lower in the *Arabidopsis* lines overexpressing *MbWRKY33*, representing the damage to plants under abiotic stress (Figure 6). The findings above imply that overexpressing *MbWRKY33* might increase the resistance of transgenic plants against intense salt stress, significantly improving how plants respond to extreme salt stress.

### 2.5. MbWRKY33 Activates Downstream Salt-Tolerance Related Genes in A. thaliana

To further clarify the regulatory mechanism of the *MbWRKY33* gene during salt stress, this experiment investigated the expression of downstream genes associated with the salt stress response, including *AtNHX1* (NM_122597.3), *AtSOS1* (NM_126259.4), *AtSOS3* (NM_122333.6), *AtRD29a* (NM_124610.3), *AtSnRK2.4* (NM_100969.4), and *AtNCED3* (NM_112304.3), which qRT-PCR detected under normal conditions. Under high-salt stress conditions, upon analyzing the expression levels of these genes, it was revealed that the overexpression lines exhibited significantly elevated expression levels relative to the WT and UL lines (Figure 7A–F). These outcomes indicate that MbWRKY33 overexpression might improve high-salt stress resistance by promoting the expression of genes related to salt tolerance.

## 3. Discussion

Abiotic stresses such as high salinity and low temperature severely affect the growth and yield of plants [35,36,37]. Nowadays, more and more studies are carried out through biotechnology to cultivate transgenic stress-resistant plants, which is beneficial to the research and breeding of horticultural crops. Although the mechanisms of action of WRKY TFs in *Malus baccata* in terms of stress resistance remain unclear, different approaches have been employed to explore the functions of many WRKY TFs in modulating plant growth, development, and reactions to abiotic stresses across various plant types [38,39,40,41]. Many WRKY transgenic families have been verified to be closely related to both biotic and abiotic stresses [42,43]. While research into the WRKY transcription factor family in other model organisms has grown comprehensively, limited research has been conducted on the stress response mechanisms controlled by WRKY transcription factors in *Malus baccata.*

Our research isolated and identified a gene *MbWRKY33* induced by salt stress from *Malus baccata*. The analysis of the gene structure and phylogeny demonstrated that this gene sequence’s full length is 1539 bp, containing 512 amino acids. Its amino acid sequence includes the WRKYGKK, a variant of all amino acid sequences of WRKY33, and the C2H2 zinc finger motif (Appendix A). The study of protein properties indicated that the theoretical molecular weight (MW) of the MbWRKY33 protein is 56775.04 kDa, the theoretical isoelectric point (pI) is 7.09, and the average hydrophilicity coefficient is -0.967, suggesting that it is a hydrophilic protein. The phylogenetic tree analysis of the MbWRKY33 protein was conducted using MEGA7.0. It was found that it has similarities with the conserved sequences of WRKY proteins from other species (Figure 1A), and it has the closest genetic relationship with RcWRKY33 (Figure 1B). the structural analysis of the MbWRKY33 protein revealed that it has the same characteristic structural domain of the WRKY family (Figure 2A) and its tertiary structure (Figure 2B), containing two conserved WRKY domains, and belongs to a member of the class I WRKY family. 

TFs are usually localized in the nucleus, where they perform the regulatory function of transcription. Numerous studies have been conducted on the subcellular localization of WRKY TFs, confirming that the fluorescence signal can only be observed in the cell nucleus. To precisely locate the exact position of the MbWRKY33 protein within the cell, determine its functional site, and understand the mechanism of action of the gene, this gene was transferred into tobacco leaves’ cells through tobacco’s transient transformation. It was found that its localization in the nucleus (Figure 3) is consistent with its role as a transcription factor [44].

Throughout the growth of plants, gene expression exhibits cell- or tissue-specificity, often shown as fluctuations in the expression levels of genes within different tissues and organs [45]. The expression of *MbWRKY33* in the new leaves, stems, roots, and mature leaves of *Malus baccata* was analyzed. Figure 4 shows that the expression levels of *MbWRKY33* in young leaves and roots are significantly higher than in stems and mature leaves, indicating that this gene is more sensitive to abiotic stresses in newly formed organs. The findings on *MbWRKY33* expression and functional characterization in *Malus baccata* align with and extend previous reports in other plant species. For instance, in *Arabidopsis thaliana*, AtWRKY33 has been shown to mediate salt tolerance by activating the expression of *RD29A* and *COR15A* [18]. Functional validation in transgenic *Arabidopsis* further highlights the conserved role of WRKY transcription factors in abiotic stress tolerance. While the overexpression of *MbWRKY33* did not alter normal growth phenotypes, transgenic lines showed a 62% higher survival rate under 200 mM NaCl than the wild type (Figure 5C). This efficacy is comparable to *AtWRKY33*-overexpressing lines, which exhibited a 55% survival rate under similar conditions [32]. Therefore, in new leaves and roots, differences in the expression changes in *MbWRKY33* induced by different stresses under abiotic stresses were observed, and it was found that *MbWRKY33* is more sensitive to high salinity, low temperature, and drought stresses. The *Arabidopsis thaliana* seeds overexpressing *MbWRKY33* confers a much higher rate of survival and enhanced growth capabilities to plants under salt-stress conditions, indicating that *MbWRKY33* can improve the tolerance of apples to high salinity and low-temperature osmotic stresses.

External stresses cause plants to rapidly release a large amount of ROS, and excessive ROS can damage plant proteins [46]. The content of H_2_O_2_ is a kind of oxidative stress signal within plant cells, which initiates the gene expression of various antioxidant enzymes and enhances the activity of antioxidant enzymes. Plant cells contain a variety of protective enzyme systems, and they mainly consist of SOD, POD, CAT, ascorbate peroxidase (APX), and glutathione reductase (GR), etc. SOD is produced through the rapid decomposition of superoxide radicals. During the formation of hydrogen peroxide, CAT, POD, and APX are mainly used to scavenge hydrogen peroxide [47]. Through the action of antioxidant enzymes, cells can indirectly prevent the generation of reactive oxygen species with vigorous oxidation activity, such as O_2_^−^, H_2_O_2_, and OH^−^, or delay the lipid peroxidation of the membrane system, guaranteeing the regular operation of various intracellular metabolic activities [48,49]. This research found that compared with the control, plants overexpressing *MbWRKY33* may experience less lipid peroxidation, membrane damage, and cell damage under salt stress conditions [50,51]. Plants are equipped with a more robust enzymatic antioxidant defense system. By scavenging ROS, it defends cells from oxidative injury (such as SOD and POD) [52]. This is consistent with the result that the overexpression of the DgWRKY5 gene in chrysanthemums enhances the tolerance to salt stress by enhancing ROS scavenging and osmotic adjustment. 

Many genes induced by ABA are expressed under high-salinity and low-temperature conditions. Significant progress in the research of the ABA-signaling pathway has shown that WRKY TFs are key nodes in the ABA-responsive signaling pathway network. Further genetic analysis indicates that WRKY proteins can directly promote or inhibit the expression of ABA-sensitive genes, thereby triggering ABA responses and affecting the expression of downstream genes [15,50]. In an attempt to analyze the possible downstream functional genes that *MbWRKY33* may act on in response to high-salt stress, we determined the expression levels of crucial genes in the high-salt stress response pathway in transgenic plants after they were treated [52,53,54,55]. The results showed that the expression level of NCED3 was increased through the ABA pathway, and the functions of SOS1, SOS3, and NHX1 were upregulated through the SOS pathway. Some studies have shown that when subjected to salt stress, the expression of the GmDREB2 gene induces the expression of downstream genes Rd29A and COR15a [56]. Furthermore, the NHX1-encoded Na^+^/H^+^ antiporter is responsible for regulating Na^+^ transport, leading to an improvement in salt tolerance. SnRK2.4 serves as a crucial regulator within the ABA-signaling pathway. After activation, it can phosphorylate many subsequent target proteins and regulate various physiological reactions to ABA [57,58,59], thus activating the salt resistance ability of plants and enhancing their salt tolerance. 

Our findings reveal that *MbWRKY33* expression is induced by salt stress in *M. baccata*, aligning with observations in other species where WRKYs act as stress-responsive regulators. For example, similarly to *AtWRKY33*’s role in *Arabidopsis* salt tolerance [18], *MbWRKY33* may modulate ion transport and osmotic adjustment via downstream target genes. This research verified the beneficial effects of *MbWRKY33* by detecting the gene activity during the ROS-scavenging process by observing the changes in the functions of SOD, POD, and CAT. Under high-salt conditions, *MbWRKY33* can be induced to participate in terrestrial plants’ abiotic stress response mechanisms. Overexpressing *MbWRKY33* enhances the resistance of plants to adverse stresses through multiple pathways, including influencing the antioxidant activity of plants and improving the tolerance of transgenic plants to high-salt stress by affecting the expression of downstream stress-related genes (Figure 8). This research involved the heterologous analysis of the function of *MbWRKY33* in *Arabidopsis thaliana.* However, whether *MbWRKY33* can perform similar functions during the ontogeny of *Malus baccata* remains to be analyzed more thoroughly. The findings suggest that *MbWRKY33* can significantly influence the salt tolerance of *Arabidopsis thaliana* and offer valuable insights for cultivating *Malus baccata* in saline–alkali land.

## 4. Materials and Methods

### 4.1. Plant Materials

The seeds of *Malus baccata* were collected from the fruits of *Malus baccata* trees outside the Horticulture Building of Northeast Agricultural University. The in vitro tissue-cultured seedlings of *Malus baccata* were propagated and rooted on MS medium and then transferred to Hoagland’s nutrient solution for acclimation culture, which lasted for 50 days. After the root system was formed, the tissue-cultured seedlings were placed in Hoagland’s nutrient solution for hydroponic culture [60]. The hydroponic solution was replaced once every 3 to 4 days. The tissue culture room was kept at approximately 25 °C, with the relative humidity between 80% and 85% [61].

### 4.2. Analysis of the Expression Pattern of MbWRKY33

For the functional analysis of *Malus baccata*, the *Malus baccata* seedlings of the same batch were grouped. Seeds of *M. baccata* were harvested from the fruits of *M. baccata* trees located outside the horticulture building of the Northeast Agricultural University. When the seedlings had grown 7 to 9 fully expanded true leaves and the root systems were well-developed, they were evenly divided into five groups for individual stress treatments. Referring to previous studies [62,63,64,65], the expression of the *MbWRKY33* gene in the roots, stems, and leaves (young leaves and mature leaves) of *Malus baccata* seedlings were analyzed. After the roots, stems, and leaves of the seedlings had matured, different parts were subjected to stresses of high salinity, low temperature, drought, and abscisic acid (ABA, Solarbio, A8060, Beijing, China) hormone, and the expression pattern of the *MbWRKY33* gene after the stress treatments were observed. The first group, which was not subjected to any stress treatment in a constant-temperature incubator, was the control group. To simulate the growth conditions of plants under high salinity, low temperature, drought, and ABA stress environments, the remaining four groups were treated with 200 mM NaCl, 4 °C, 6% PEG6000, and 100 µM ABA, respectively. Before autoclaving, 200 mM NaCl, 6% PEG6000, and 100 µM ABA were added to solid 1/2 MS medium, and seedlings were transplanted onto the treated medium [66,67,68,69]. Samples were taken from different parts at the seven time points shown in Figure 4. After sampling, liquid nitrogen (N_2_) was employed to freeze the sample rapidly and then stored in a refrigerator at −80 °C.

### 4.3. RNA Extraction and Cloning of MbWRKY33

Tender young leaves, roots, stems, and mature leaves of *Malus baccata* seedlings were selected as experimental materials for extracting total RNA. Subsequently, the RNA was purified using the Universal Plant Total RNA Isolation Kit from Vazyme (Nanjing, China). Using the RNA template, the first-strand cDNA was synthesized with TransScript First Strand cDNA Synthesis Super Mix (TransGen Biotech, Beijing, China). In addition, a pair of specific primers (*MbWRKY33*-F/R; Appendix A) was designed. The target gene was amplified using a PCR instrument [70]. Subsequently, the PCR product was ligated into the T5 cloning vector, and the sample was then submitted for sequencing. 

### 4.4. Subcellular Localization Analysis of MbWRKY33

The subcellular localization expression vector was constructed in tobacco leaves using the pCAMBIA1300-GFP plasmid (Beyotime Biotechnology, Shanghai, China) and the full-length sequence of the *MbWRKY33* gene. Specific primers were designed using the restriction enzyme cutting sites of *Sal*I and *BamH*I (*MbWRKY33*-sl F/sl R; Appendix A), and *MbWRKY33* with restriction enzyme cutting sites was synthesized by PCR amplification [71,72,73,74]. Both the PCR product and the 35S-sGFP-pCAMBIA1300 vector were digested with double enzymes, and then the target fragment was ligated with the vector to construct the *MbWRKY33*-GFP transient expression vector [75]. Finally, the transformation of *Agrobacterium tumefaciens* GV3101 was carried out. After the bacterial solution was activated overnight, the bacterial cells were collected and resuspended in a buffer solution (10 mM MgCl_2_ + 10 mM MES + 200 μM acetosyringone), and modified the OD value was modified to 0.4. After that, it remained stationary at ambient temperature for 2–3 hours [76,77]. The infection solution was injected into the fully expanded leaves of Nicotiana benthamiana that had been growing for 5–6 weeks using a syringe [78,79], after culturing the leaves under low light for two days. The stained leaves were placed under a laser confocal microscope to observe the position of the protein encoded by MbWRKY33 in the cells.

### 4.5. Sequence Alignment and Structure Prediction of MbWRKY33

DNAMAN 6.0 software was employed to translate the sequence and predict the primary structure of *MbWRKY33*. The Expasy-Prot Param tool at https://web.expasy.org/protparam/ (accessed on 10 March 2025) was applied to predict the primary structure of the MbWRKY33 protein, which involves determining the relative molecular mass and the theoretical isoelectric point, instability coefficient, and the Grand average of hydropathicity [80,81]. The amino acid sequences of the MbWRKY33 protein, as well as those of WRKY proteins from other plants, were retrieved from NCBI (https://www.ncbi.nlm.nih.gov/, accessed on 5 February 2025); after that, a phylogenetic tree was created using MEGA7.0. DNAMAN6.0 was also used to perform multiple sequence alignments.

The secondary structure of the MbWRKY33 protein was predicted using SOPMA (https://npsa.lyon.inserm.fr/cgi-bin/npsa_automat.pl?page=/NPSA/npsa_sopma.html, accessed on 6 March 2025). The structural domain of the MbWRKY33 protein was predicted in the SMART database (http://smart.embl-heidelberg.de/, accessed on 7 March 2025). Its tertiary structure was predicted using SWISS-MODEL (https://swissmodel.expasy.org/, accessed on 9 March 2025).

### 4.6. Expression Analysis of the MbWRKY33 Gene

The expression analysis of *MbWRKY33* was used to evaluate the degree of being subjected to various abiotic stresses in different organs. RNA was extracted from 0.1 g of each tissue organ (roots, stems, new leaves, old leaves) of *Malus baccata*, reverse-transcribed into cDNA, and used as a template for the RT-qPCR analysis. We designed the qPCR primers (*MbWRKY33*-qF/qR; Appendix A). The qPCR reaction system was established according to the method proposed by Li et al. [82,83]. The internal reference gene selected in this study was *MbActin* (*Malus baccata*), the corresponding primers (*MbActin*-qF/qR; Appendix A) were designed, and this gene showed consistent expression. The 2^−∆∆Ct^ method was used to detect the relative expression level of the *MbWRKY33* gene [84]. Technical replicates were carried out by repeating the experiment three times.

### 4.7. Obtaining Transgenic Arabidopsis thaliana Lines

The 5’-end (*Sal*I restriction site) and 3’-end (*BamH*I site) of the *MbWRKY33* cDNA were amplified using the primers *MbWRKY33*-F and *MbWRKY33*-R. It was then ligated into the pCAMBIA1300 vector, and the *MbWRKY33*-OE vector was constructed using homologous recombination primers and techniques. The overexpression vector (*MbWRKY33*-pCAMBIA1300) and the empty vector (pCAMBIA1300) were transformed into *Agrobacterium rhizogenes,* respectively, and then the bacterial solution was transferred into *Arabidopsis thaliana* through a floral dip method [85]. After that, the positive transgenic lines (T1 generation seeds) were sown on the screening medium supplemented with 50 mg/L kanamycin. When the *Arabidopsis thaliana* plants matured, the T2 generation seeds were collected and continued to be cultured. Six T2 generation *Arabidopsis thaliana* plants were randomly selected to extract leaf DNA, and three plants exhibiting relatively high expression levels were screened for the T3 generation. Subsequently, after planting the T3 generation plants, high-expression transgenic lines were chosen from this generation (S1, S3, S4) to produce pure T3 generation plants. The chosen T3 generation lines were utilized for subsequent analysis, with wild-type (WT) and untransformed (UL) controls established.

### 4.8. High-Salt Stress Treatment

Wild-type (WT), empty vector (UL), and T3 generation transgenic *Arabidopsis thaliana* lines (S1, S3, S4) were used as experimental materials. They were cultured in a light incubator for 4 weeks with a cycle of 16 hours of light and 8 hours of darkness. Twenty well-grown strains were selected, and every 4 strains were planted in the same pot for salt stress (200 mM NaCl) treatment [86]. For 7 consecutive days, daily irrigation with the salt solution was carried out for the WT, UL, and T3 generation transgenic lines. Then, the seedlings were watered with clear water for one week to remove the residual NaCl solution [41,87]. During each treatment, the corresponding marks were made for each row, and different conditions of the plants were noted down. In addition, leaf samples of *Arabidopsis thaliana* from the wild type (WT), empty vector (UL), and transgenic lines (S1, S3, S4) were collected for future use [88,89].

### 4.9. Measurement of Physiological Indices

Subsequently, the *Arabidopsis thaliana* lines (WT, UL, S1, S3, S4) were grown in a soil mixture consisting of peat, vermiculite, and perlite at a ratio of 1:2:1. And the survival and physiological parameters of the leaves of the above-collected *Arabidopsis thaliana* lines were measured. The chlorophyll content was assayed through the extraction technique following Zhang’s method [90]. According to the methods of Li Zhongguang et al. [63], we measured the activities of SOD, POD, MDA, and CAT through the nitroblue tetrazolium photoreduction method, guaiacol method, thiobarbituric acid (TBA) method, and ultraviolet absorption method, respectively. The contents of H_2_O_2_ and O_2_^−^ were measured using diaminobenzidine (DAB) and nitroblue tetrazolium (NBT), respectively [91]. According to the guidance on postharvest physiological and biochemical experiments of fruits and vegetables compiled by Cao Jiankang et al., we employed the air-extraction method to measure the relative electrical conductivity and the sulfosalicylic acid method to assay the proline content [92].

### 4.10. Expression Analysis of Salt Stress-Related Genes in Arabidopsis thaliana Lines Overexpressing MbWRKY33

In total, 0.1. g of *Arabidopsis thaliana* leaves were taken from each line. These leaves were used for RNA extraction. RNA was extracted, and using the first strand as a template, WT, UL, and transgenic *Arabidopsis thaliana* (S1, S3, S4) were transformed into cDNA after being treated under high-salt, low-temperature, and normal conditions. Using MbActin as the internal reference gene, RT-qPCR was used to detect the related downstream genes of *MbWRKY33* in WT, UL, S1, S3, and S4, including salt-stress-related genes (*AtNHX1, AtSOS1, AtSOS3, AtRD29a, AtSnRK2.4,* and *AtNCED3*), to explore the expression levels of genes associated with abiotic stress response [93]. The reaction system referred to the above qPCR system of *MbWRKY33*, and the 2^−∆∆Ct^ method was used to estimate the expression level of the target gene [94].

### 4.11. Statistical Analysis

Each sample in this study was subjected to three biological replicates, and the average value of the repeated experiments was used as the value of the corresponding sample [95]. The experimental results were expressed as the mean value and standard error (SE) [96]. A one-way analysis of variance (ANOVA) was performed using SPSS 26.0 software to determine the statistical differences (* *p* ≤ 0.05, ** *p* ≤ 0.01) [97]. 

## 5. Conclusions

In this research, we cloned and identified the WRKY transcription factor gene *MbWRKY33* from *Malus baccata* and found that it was located in the nucleus through subcellular localization. According to the phylogenetic tree analysis, *MbWRKY33* has the closest genetic relationship with RcWRKY33. *MbWRKY33* has higher expression levels in young leaves and roots, showing a greater response to high-salinity and low-temperature signals. Moreover, we studied the expression levels and related physiological indices of *Arabidopsis thaliana* overexpressing *MbWRKY33* under normal and abiotic stress conditions, which indicated that this gene enhanced the tolerance to high-salinity and low-temperature conditions; *MbWRKY33* is highly sensitive to high salinity. Similarly, *MbWRKY33* may influence the plant’s response to stress by controlling antioxidant activity and the expression of genes associated with downstream disease resistance. To sum up, the research results suggest that the overexpression of *MbWRKY33* may be involved in regulating the tolerance of plants to abiotic stress, which provides a theoretical basis for the molecular breeding improvement of *Malus baccata*.

## Figures and Tables

**Figure 1 ijms-26-05833-f001:**
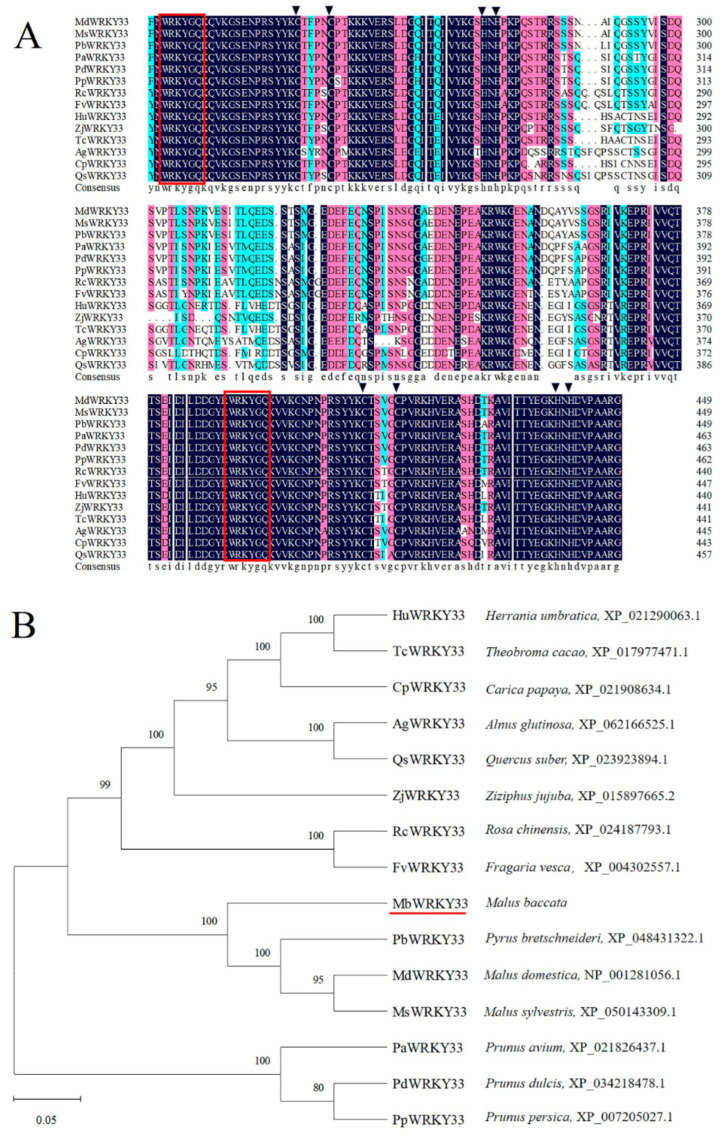
Alignment and phylogenetic tree analysis of other species’ MbWRKY33 and WRKY amino acid sequence. (**A**) MbWRKY33 in comparison to other species’ WRKY33 proteins. Red boxes indicate two WRKY conserved structural domains; the black triangle marks the acetylene zinc finger motif of WRKY. (**B**) WRKY33 and MbWRKY33 protein phylogenetic trees of different species. The red underline shows the protein MbWRKY33. The numerical values represent the genetic relatedness among other species, and the phylogenetic tree was created on MEGA7.0.

**Figure 2 ijms-26-05833-f002:**
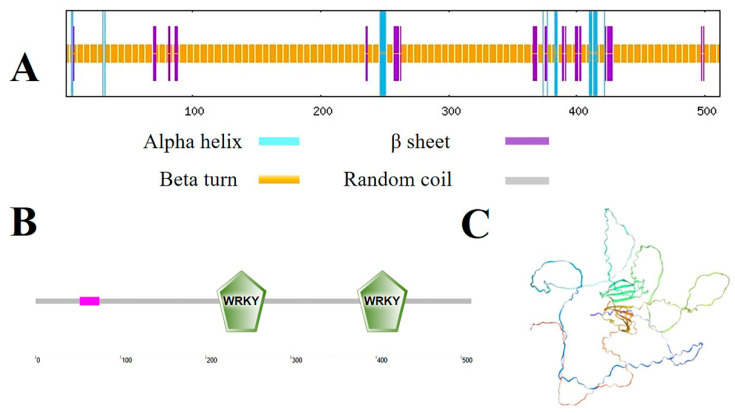
Secondary structure and tertiary structure prediction of MbWRKY33 protein. (**A**) MbWRKY33’s structural analysis at the secondary level. (**B**) Conserved structural domain study of MbWRKY33. (**C**) Tertiary structural prediction of MbWRKY33.

**Figure 3 ijms-26-05833-f003:**
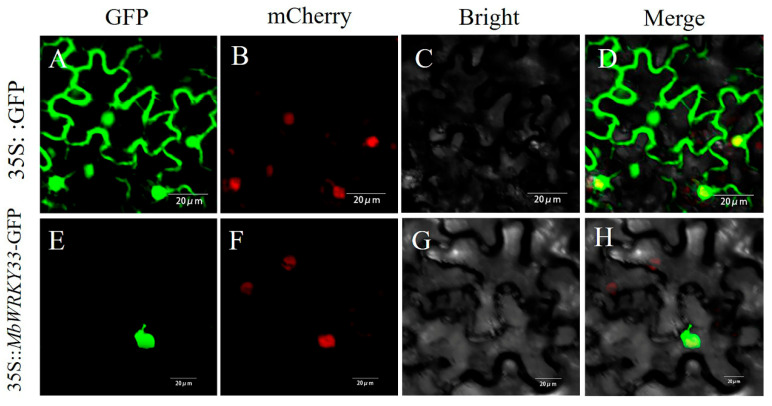
Subcellular localization of MbWRKY33 protein. 35S::*MbWRKY33*::GFP was expressed transiently into tobacco leaves with 35Spro::GFP as a positive control. (**A**,**E**) GFP signals; (**B**,**F**) mCherry; (**C**,**G**) Bright field; (**D**,**H**) Merge. mCherry as a nuclear marker. Yellow indicates GFP and mCherry colocalization. Scale bars correspond to 20 µm.

**Figure 4 ijms-26-05833-f004:**
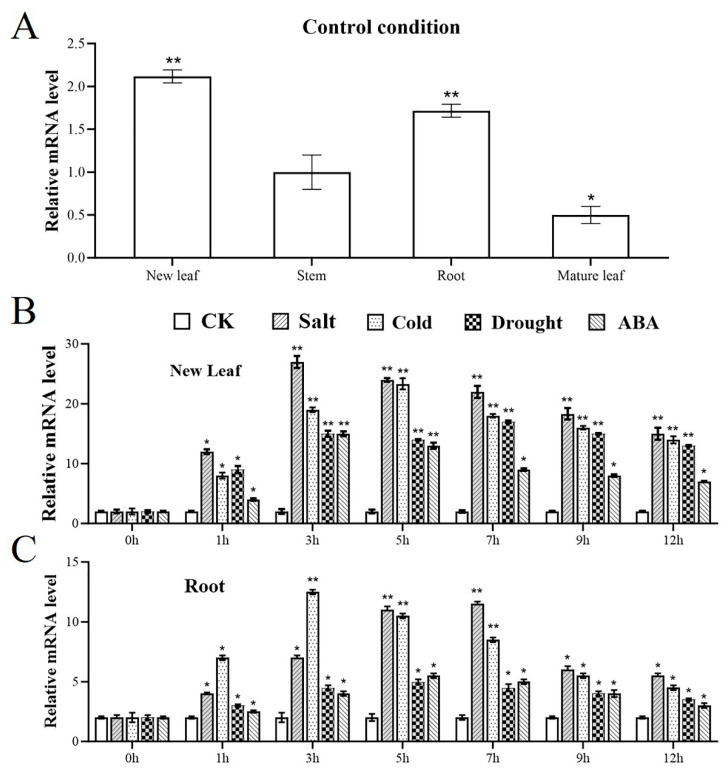
Tissue-specific and stress-responsive expression patterns of *MbWRKY33* in *Malus baccata.* (**A**) New leaves, stems, roots, and mature leaves all exhibit varying degrees of *MbWRKY33* expression. (**B**) Time-course of *MbWRKY33* expression in new leaves in control and under salt (200 mM NaCI), low-temperature (4 °C), dehydration (6% PEG6000), and abscisic acid (50 µM ABA) treatments. (**C**) Time-course of *MbWRKY33* expression in roots in control and under salt (200 mM NaCl), low-temperature (4 °C), dehydration (6% PEG6000), and abscisic acid treatments (50 µM ABA). Error bars indicate the standard deviation. Asterisks above the error bars indicate a significant difference between the treatment and control (Student’s *t*-test; * *p* ≤ 0.05, ** *p* ≤ 0.01).

**Figure 5 ijms-26-05833-f005:**
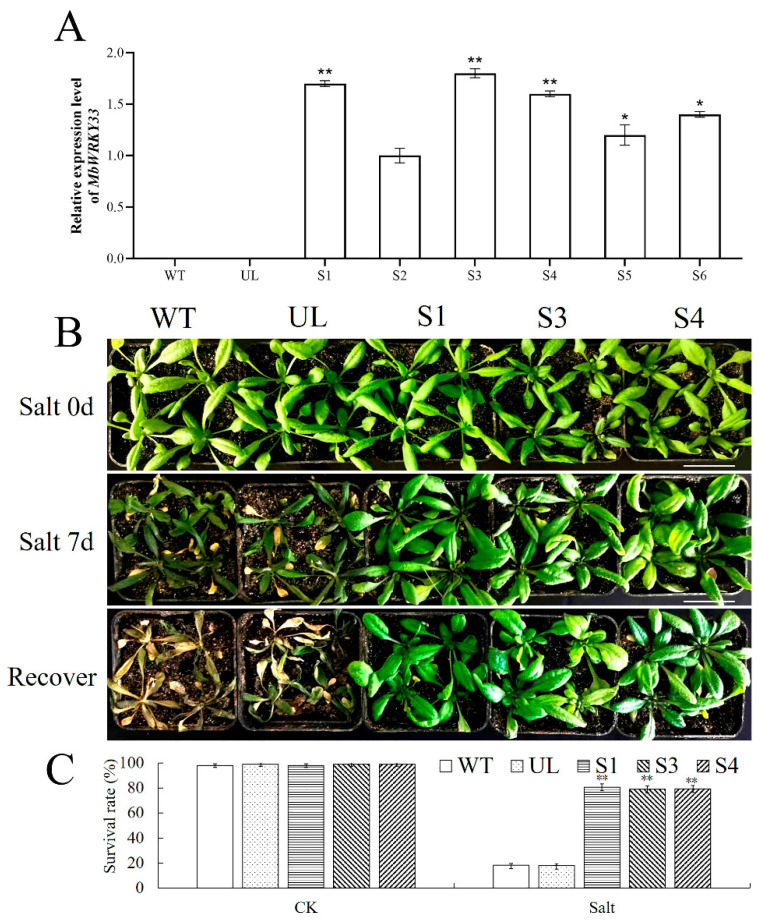
*MbWRKY33* confers salt tolerance in *Arabidopsis*. (**A**) Expression of the *MbWRKY33* transcript in transgenic lines S1–S6, null-loaded line UL, and wild-type WT. (**B**) Transgenic high-expression lines (S1, S3, and S4), UL, and WT phenotypes. Scale bar is 3 cm. (**C**) Survival of each *Arabidopsis* line under high-salt stress. The asterisks above indicate statistically significant comparisons (* *p* ≤ 0.05; ** *p* < 0.01) with WT differences.

**Figure 6 ijms-26-05833-f006:**
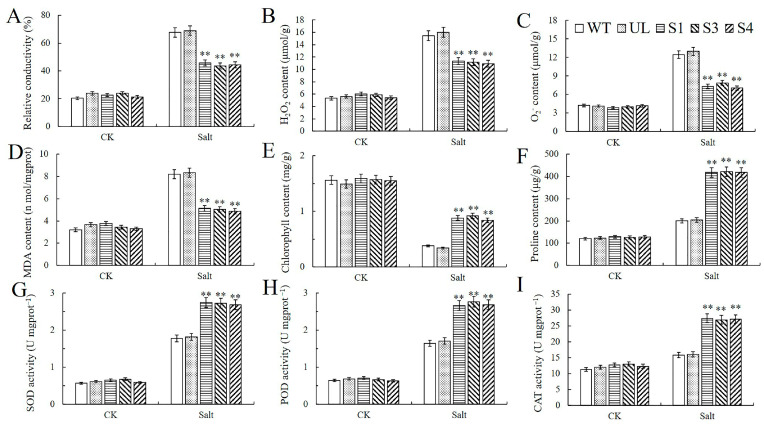
Effects of *MbWRKY33* on stress-related physiological indices in *Arabidopsis* under high-salt stress. Contents of (**A**) Relative conductivity, (**B**) H_2_O_2_, and (**C**) O_2_^−^, (**D**) MDA, (**E**) chlorophyll, (**F**) proline, and the activities of (**G**) SOD, (**H**) CAT, and (**I**) POD in the WT, UL, and *MbWRKY33*-OE lines (S1, S3, and S4) under 200 mM NaCl treatment for 7 days. The mean of three replicates is the standard error. The transgenic lines differed significantly from the WT lines, as indicated by the asterisk above the error bars (Student’s *t*-test, ** *p* ≤ 0.01).

**Figure 7 ijms-26-05833-f007:**
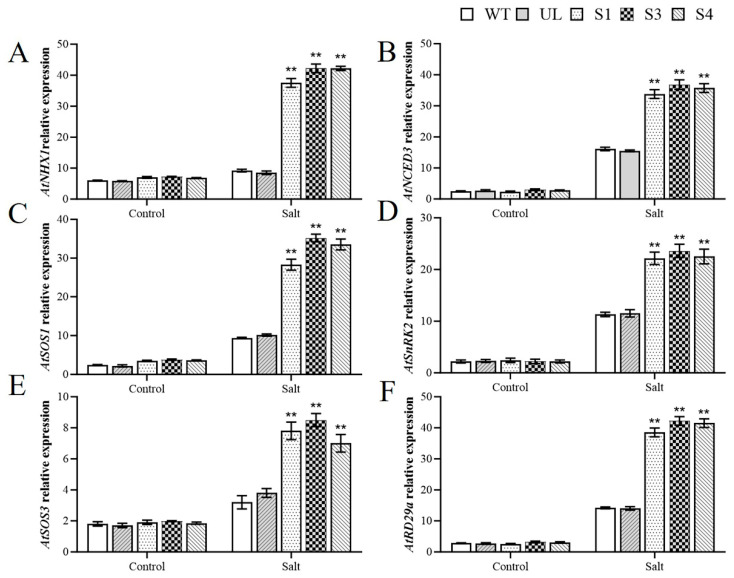
Effects of *MbWRKY33* on the expression of genes associated with salt tolerance in *Arabidopsis* under high-salt stress. (**A**) *AtNHX1*; (**B**) *AtNCED3*; (**C**) *AtSOS1*; (**D**) *AtSnRK2*; (**E**) *AtSOS3;* and (**F**) *AtRD29a*. The mean of three duplicate experiments is the standard error. The transgenic lines differed significantly from the WT lines, as indicated by the asterisk above the error bars (Student’s *t*-test, ** *p* ≤ 0.01).

**Figure 8 ijms-26-05833-f008:**
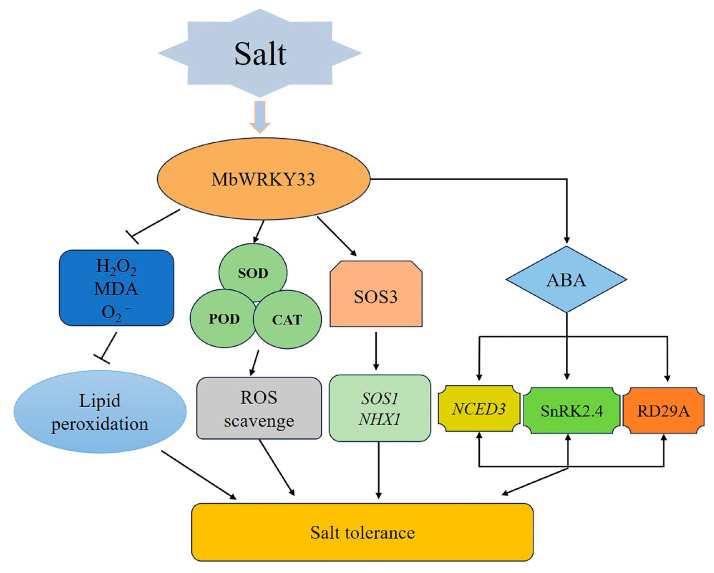
A possible model for the role of *MbWRKY33* during plant resistance to salt stress. The activated *MbWRKY33* substantially reduced cellular damage by reactive oxygen species and lipid peroxides, promoting the upregulated expression of stress-related genes. *MbWRKY33* receives signals to be activated when plants are exposed to a high-salt environment. On the one hand, *MbWRKY33* participates in the SOS pathway and binds to the promoters of *AtSOS3* to activate the expression of the downstream genes, *AtSOS1* and *AtNHX1*, to enhance the plant’s tolerance to the high-salt environment. On the other hand, *MbWRKY33* participates in the ABA-signaling pathway and promotes the expression of the downstream genes *AtRD29a*, *AtNCED3,* and *AtSnRK2* to realize the enhancement in salt tolerance in plants.

## Data Availability

Data will be made available on request.

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
