# Peer review of "Overexpression of a *Malus baccata* (L.) Borkh WRKY Factor Gene *MbWRKY33* Increased High Salinity Stress Tolerance in *Arabidopsis thaliana"

_ijms, 2025, doi:10.3390/ijms26125833_

Round 1
Reviewer 1 Report
Comments and Suggestions for Authors
Main Question Addressed:
This study investigates whether overexpression of the MbWRKY33 gene from Malus baccata enhances salt stress tolerance in Arabidopsis thaliana. Specifically, the authors explore the gene’s localization, expression patterns under various abiotic stresses, and its functional roles in transgenic lines.
Originality and Relevance to the Field:
The topic is timely and relevant. As climate change intensifies, developing crops with enhanced stress tolerance is a major goal in plant biotechnology. WRKY transcription factors are known regulators in stress responses, but functional insights into MbWRKY33 particularly from Malus baccata, a cold- and salt-tolerant wild apple species—remain limited. This research addresses that knowledge gap and adds valuable data to the field of stress physiology and molecular breeding. The study also offers potential for improving salinity tolerance in apple cultivars and other crops.
Contribution to the Field:
- Functional validation of a Malus baccata WRKY TF under salt stress.
- Demonstration of its nuclear localization and strong stress-responsive expression in young leaves and roots.
- Evidence of enhanced antioxidant enzyme activity and reduced oxidative damage in overexpressing lines.
- Upregulation of key stress-related genes (e.g., AtNHX1, AtRD29a, AtSOS1, and AtSnRK2.4), suggesting involvement in ABA and SOS pathways.
Methodological Evaluation:
The methods are generally robust and appropriate for the research question. The cloning, subcellular localization, qRT-PCR expression profiling, and physiological assessments are well executed. However, for future improvement, the authors may consider:
- Including western blot or GUS staining to validate the expression of MbWRKY33 at the protein level.
- Providing more detailed statistics in some of the figures, such as sample sizes and exact p-values.
- Expanding the phenotype analysis of transgenic lines beyond 7 days to assess long-term tolerance and recovery under salt stress.
Overall Assessment:
This is a scientifically sound and well-executed study that addresses a significant gap in stress tolerance mechanisms in Malus baccata. The research provides important functional evidence for MbWRKY33 as a candidate gene for improving salt stress resistance through transcriptional regulation and ROS scavenging. The manuscript is suitable for publication after minor improvements and adds value to the molecular stress physiology field.
Comments on the Quality of English LanguageThe manuscript presents valuable insights, the overall clarity and readability would benefit from careful English language and grammar editing.
Author Response
- Methodological Evaluation:
The methods are generally robust and appropriate for the research question. The cloning, subcellular localization, qRT-PCR expression profiling, and physiological assessments are well executed. However, for future improvement, the authors may consider:
Including western blot or GUS staining to validate the expression of MbWRKY33 at the protein level.
Providing more detailed statistics in some of the figures, such as sample sizes and exact p-values.
Expanding the phenotype analysis of transgenic lines beyond 7 days to assess long-term tolerance and recovery under salt stress.
Response: Thank you for your valuable feedback on our manuscript. We will subsequently validate the expression of MbWRKY33 at the protein level, and then expand the phenotypic analysis of transgenic lines beyond 7 days to assess long-term tolerance and recovery under salt stress. Thank you for your suggestions.

Reviewer 2 Report
Comments and Suggestions for Authors
This manuscript presents a study on the functional characterization of the WRKY transcription factor MbWRKY33 from Malus baccata and its role in conferring salt stress tolerance in Arabidopsis thaliana. The topic is relevant and potentially impactful in the field of plant stress physiology and genetic improvement. The authors employed standard molecular and physiological assays to support their findings However, the manuscript in its current form exhibits several issues concerning both linguistic clarity and scientific rigor, which should be thoroughly addressed to improve its overall quality and impact. A major revision is required to improve the manuscript’s readability and scientific rigor.
Major Comments:
- The manuscript does not provide a clear rationale for selecting MbWRKY33 as the target gene for functional characterization. Given that the WRKY transcription factor family contains a large number of members, it is essential to explain why MbWRKY33 was chosen among them. Was its expression strongly induced under specific stress conditions in preliminary experiments? Has its homolog in other species been previously reported to be involved in stress responses? If so, such evidence should be included and discussed in the Introduction to better support the gene selection and highlight the novelty of the study.
- The transgenic Arabidopsis thaliana lines overexpressing MbWRKY33 were only verified at the mRNA level using qRT-PCR. To ensure the reliability of the transgenic system and the observed phenotypes, further validation at the genomic level (e.g., genomic PCR to confirm insertion) and protein level (e.g., western blot or GFP fusion protein detection) is recommended. In addition, it is unclear whether WRKY33 has homologs in Arabidopsis, and if so, whether the overexpression construct was designed to avoid sequence similarity that might interfere with endogenous WRKY gene expression. The authors should clarify whether potential off-target effects were evaluated or minimized, particularly if homologous regions exist. From the figure, it appears that homologous sequences may have been avoided, but a brief explanation and supporting alignment would help strengthen this point.
- The manuscript lacks sufficient detail regarding how the abiotic stress treatments, particularly abscisic acid (ABA), were applied to Malus baccata seedlings (Lines 400-404). It is stated that the expression of MbWRKY33 was analyzed after stress treatment, but the precise method of ABA application—such as whether it was applied via foliar spray, root drenching, or added to the growth medium—is not specified.
Minor Comments:
- In the reference list, Latin names of plant species should be italicized. Please carefully review the entire manuscript to ensure all Latin names are correctly formatted in italics (line 599). In addition, there is inconsistency in the capitalization of words in reference titles—for instance, words in the titles from Lines 559–560 are capitalized, while those in Lines 557–558 are not. Reference 31 is highlighted in red, please modify it. These inconsistencies indicate that the reference formatting requires thorough revision. Please carefully check all references and revise them according to the journal’s formatting requirements.
- Figure 1 primarily displays the nucleotide and deduced amino acid sequence of MbWRKY33, which is informative but not essential to the main flow of the manuscript. This figure could be moved to the supplementary material to improve readability and streamline the main text. A brief reference to it in the Results section would still allow interested readers to access it if needed.
The overall quality of the English language in this article is generally well-written, yet there are certain aspects that necessitate refinement.
- Line 167, “nvestigate” → “investigate”
- Line 135, “witn” → “with”
Please meticulously review the entire text for the accuracy of English tenses and the fluidity of the language and proceed with the necessary revisions.
Author Response
Dear Editor and Reviewers:
Thank you for your letter and comments concerning our manuscript. Those comments are all valuable and very helpful for revising and improving our paper, as well as having important guiding significance for our research. We have studied the comments carefully and have made corrections, which we hope will meet with approval. The main corrections in the paper and the responses to comments are listed below. The reviewer's comments are listed below in italics. Our responses are given in bold, and changes to the manuscript are given in red text.
Responds to the reviewer's comments:
Reviewers' comments:
Major Comments:
1.The manuscript does not provide a clear rationale for selecting MbWRKY33 as the target gene for functional characterization. Given that the WRKY transcription factor family contains a large number of members, it is essential to explain why MbWRKY33 was chosen among them. Was its expression strongly induced under specific stress conditions in preliminary experiments? Has its homolog in other species been previously reported to be involved in stress responses? If so, such evidence should be included and discussed in the Introduction to better support the gene selection and highlight the novelty of the study.
Response: Thank you for pointing out the issue in our manuscript. We have supplemented clear rationales for selecting MbWRKY33 as the target gene for functional characterization in the Introduction and Discussion sections, with references provided in lines 105-114 and 342-345 of the manuscript.
- The transgenic Arabidopsis thaliana lines overexpressing MbWRKY33 were only verified at the mRNA level using qRT-PCR. To ensure the reliability of the transgenic system and the observed phenotypes, further validation at the genomic level (e.g., genomic PCR to confirm insertion) and protein level (e.g., western blot or GFP fusion protein detection) is recommended. In addition, it is unclear whether WRKY33 has homologs in Arabidopsis, and if so, whether the overexpression construct was designed to avoid sequence similarity that might interfere with endogenous WRKY gene expression. The authors should clarify whether potential off-target effects were evaluated or minimized, particularly if homologous regions exist. From the figure, it appears that homologous sequences may have been avoided, but a brief explanation and supporting alignment would help strengthen this point.
Response: Thank you for pointing out the issue in our manuscript. We have validated the reliability of the overexpressing Arabidopsis thaliana lines at the genomic level in lines 206-208.
- The manuscript lacks sufficient detail regarding how the abiotic stress treatments, particularly abscisic acid (ABA), were applied to Malus baccata seedlings (Lines 400-404). It is stated that the expression of MbWRKY33 was analyzed after stress treatment, but the precise method of ABA application—such as whether it was applied via foliar spray, root drenching, or added to the growth medium—is not specified.
Response: Thank you for pointing out the issue in our manuscript. We have supplemented the application details of abiotic stress treatments such as ABA in lines 390-392.
Minor Comments:
- In the reference list, Latin names of plant species should be italicized. Please carefully review the entire manuscript to ensure all Latin names are correctly formatted in italics (line 599). In addition, there is inconsistency in the capitalization of words in reference titles—for instance, words in the titles from Lines 559–560 are capitalized, while those in Lines 557–558 are not. Reference 31 is highlighted in red, please modify it. These inconsistencies indicate that the reference formatting requires thorough revision. Please carefully check all references and revise them according to the journal’s formatting requirements.
Response: Thank you for pointing out the issue in our manuscript. We have carefully checked all the references and modified them according to the formatting requirements of the journal.
- Figure 1 primarily displays the nucleotide and deduced amino acid sequence of MbWRKY33, which is informative but not essential to the main flow of the manuscript. This figure could be moved to the supplementary material to improve readability and streamline the main text. A brief reference to it in the Results section would still allow interested readers to access it if needed.
Response: Thank you for your valuable feedback on our manuscript. We have moved Figure 1 to the supplementary materials to enhance readability and simplify the main text.
Comments on the Quality of English Language
The overall quality of the English language in this article is generally well-written, yet there are certain aspects that necessitate refinement.
- Line 167, “nvestigate” → “investigate”
Response: We are very sorry about our mistake. We have changed them. The changes are in line 159.
- Line 135, “witn” → “with”
Response: We are very sorry about our mistake. We have changed them. The changes are in line 129.
- Please meticulously review the entire text for the accuracy of English tenses and the fluidity of the language and proceed with the necessary revisions.
Response: Thank you for your valuable feedback on our manuscript. We have carefully reviewed the entire text, checked the accuracy of English tenses and the fluency of the language, and made necessary revisions.

Reviewer 3 Report
Comments and Suggestions for Authors
The authors can improve the captions of all the figures by including more details and highlighting major observations. For example, figure 1 is missing information about the red boxes. In the Methods section, a bioinformatics analysis should be included, and the initial analysis of the putative gene should be explained in detail in the Results section.
Comments on the Quality of English LanguageCan be improve
Author Response
Comments and Suggestions for Authors
The authors can improve the captions of all the figures by including more details and highlighting major observations. For example, figure 1 is missing information about the red boxes. In the Methods section, a bioinformatics analysis should be included, and the initial analysis of the putative gene should be explained in detail in the Results section.
Response: Thank you for pointing out the issue in our manuscript. We have improved the captions of all figures by supplementing details and highlighting major findings. Additionally, bioinformatics analysis has been incorporated into the Methods section, and the Results section now includes an expanded explanation of the preliminary analysis of the putative genes.

Reviewer 4 Report
Comments and Suggestions for Authors
The paper reports the analysis of MbWRKY33 on abiotic stress. The authors describe the features of MbWRKY33, including the domains and crucial residues of WRKY transcription factors, as well as its localization and functional analysis in Arabidopsis using overexpression approaches. Validating the role of MbWRLY33 in salt stress using RT-qPCR and phenotyping of overexpression lines in Arabidopsis. The paper is at some extent organized but is weak in distinguishing between the sampling tissues in Malus baccata and Arabidopsis thaliana. It is suggested to improve the introduction section by adding information on the role of WRKYs in abiotic stress in other species. Improve the discussion section by clarifying the impact of the data in the context of previously described WRKYs.
Lines 59-61. Clarifying “functional genes” is required. In principle, most genes can function in plant biology.
Lines 38-67. It should be useful to organize the ideas. Particularly, the information displayed in lines 43-57. It seems that lines 40-43 and 57-59 have been repeated.
Lines 70-72. References are required.
Lines 98-99. Specification of the species that corresponds to CsERKY2 is required.
Lines 113-117. Additional information can facilitate the importance of the MbWRKY33 as a subject of study. Is it a gene identified as an altered gene in RNA-seq data? or another feature?, providing additional information should be helpful.
Lines 129-130. Clarification on the sequencing of MbWRKY33. It seems that sequencing is supporting the confidence of working with MbWRKY33. This should be noted in the manuscript because the sequence for MbWRKY33 is already available on the public database.
Figure 1, legends. Indication of the square highlighted residues can be beneficial.
There is a lack of information in the methodology section regarding searches for WRLY33s in other species.
In Figure 2, legends, it is necessary to indicate what the number means on the nodes of phylogeny. It is recommended to specify the software used to create the phylogeny.
Lines 181-192. It is not clear if expression analysis was performed for mature leaves and stems under stress, as there is no data displayed on Figure 5 for these plant tissues.
Lines 302-304. References are required.
Lines 309-324. Improve the discussion of these data by discussing how the new data impacts the previously reported data, at least in other species.
Lines 342-344. References are required.
Lines 363-385. I am not sure if the model presented is fully supported. Upgrading the model using the main findings would be beneficial.
Lines 397-400. Specifying the age of the seedling should be valuable.
Lines 405-411. Specifying the number of days the plants were exposed to the different treatment is required. Also, recovery experiments are required. Sampling specification is required since Figure 4 displays localization data instead of tissue samples.
Lines 414-415. Specifying sampling should be useful.
Section 4.5. All software use needs to be cited.
Section 4.6. Specification of tissue sampling is required. Also, it is required to specify the amount of tissue used. Extended information on generating and processing RNA-qPCR data is required.
Lines 467-468. Specification of sequence information for primer is required.
Line 470-473. Verification that “Agrobacterium rhizogenes” is correct for the experiment mentioned.
Lines 486-488. Specifying the soil material used is valuable.
Section 4.10. The sampling of tissue requires specific specifications. Extended information on generating and processing RNA-qPCR data is required.
Consistency in the format of letters in the panels of the figure is recommended. See figure 3 compared to the other figures.
Consistency in the mentioned fusion of WRKY33 with GFP is required, for instance, 35S::MbWRKY33::GFP versus 35Spro::MbWRKY33::GFP.
It is suggested to have a clearly defined definition for sampling, data acquisition, and processing during RT-qPCR, and for each species. Determining the new and mature leaves.
The manuscript has typos. For instance, use of cursive genes (for instance line 96-98, 100-101). 113-117. Use cursive for species, for instance, line 435. Line 442.
Author Response
Dear Editor and Reviewers:
Thank you for your letter and comments concerning our manuscript. Those comments are all valuable and very helpful for revising and improving our paper, as well as having important guiding significance for our research. We have studied the comments carefully and have made corrections, which we hope will meet with approval. The main corrections in the paper and the responses to comments are listed below. The reviewer's comments are listed below in italics. Our responses are given in bold, and changes to the manuscript are given in red text.
Responds to the reviewer's comments:
Reviewers' comments:
Reviewer #4:
Comments and Suggestions for Authors
The paper reports the analysis of MbWRKY33 on abiotic stress. The authors describe the features of MbWRKY33, including the domains and crucial residues of WRKY transcription factors, as well as its localization and functional analysis in Arabidopsis using overexpression approaches. Validating the role of MbWRLY33 in salt stress using RT-qPCR and phenotyping of overexpression lines in Arabidopsis.
- The paper is at some extent organized but is weak in distinguishing between the sampling tissues in Malus baccata and Arabidopsis thaliana. It is suggested to improve the introduction section by adding information on the role of WRKYs in abiotic stress in other species. Improve the discussion section by clarifying the impact of the data in the context of previously described WRKYs.
Response: Thank you for pointing out the issue in our manuscript. We have improved the introduction section by adding information on the role of WRKYs in abiotic stress in other species in lines 40-51. We have improved the discussion section in lines 342-345.
- Lines 59-61. Clarifying “functional genes” is required. In principle, most genes can function in plant biology.
Response: Thank you for pointing out the issue in our manuscript. The “functional genes” have been re-clarified in lines 53-54.
- Lines 38-67. It should be useful to organize the ideas. Particularly, the information displayed in lines 43-57. It seems that lines 40-43 and 57-59 have been repeated.
Response: Thank you for pointing out the issue in our manuscript. We have reorganized the ideas in lines 40-51 and deleted the repetitive content in lines 57-59.
- Lines 70-72. References are required.
Response: Thank you for your valuable feedback on our manuscript. We have added the corresponding references.
- Lines 98-99. Specification of the species that corresponds to CsERKY2 is required.
Response: Thank you for your valuable feedback on our manuscript. We have added the species corresponding to CsERKY2 in lines 90-91.
- Lines 113-117. Additional information can facilitate the importance of the MbWRKY33 as a subject of study. Is it a gene identified as an altered gene in RNA-seq data? or another feature?, providing additional information should be helpful.
Response: Thank you for your valuable feedback on our manuscript. We have supplemented the elaboration on the importance of MbWRKY33 as the research subject in lines 105-114.
- Lines 129-130. Clarification on the sequencing of MbWRKY33. It seems that sequencing is supporting the confidence of working with MbWRKY33. This should be noted in the manuscript because the sequence for MbWRKY33 is already available on the public database.
Response: Thank you for your valuable feedback on our manuscript. The sequence number of MbWRKY33 has not been publicly available in the public database, so it is not labeled in the manuscript.
- Figure 1, legends. Indication of the square highlighted residues can be beneficial.
Response: Thank you for your valuable feedback on our manuscript. We have marked the square-highlighted residues in Legend S1.
- There is a lack of information in the methodology section regarding searches for WRKY33s in other species.
Response: Thank you for your valuable feedback on our manuscript. We have added information about WRKY33 in other species to the introduction section in lines 105-114.
- In Figure 2, legends, it is necessary to indicate what the number means on the nodes of phylogeny. It is recommended to specify the software used to create the phylogeny.
Response: Thank you for your valuable feedback on our manuscript. We have added the corresponding information in Figure 1 legend in lines 143-144.
- Lines 181-192. It is not clear if expression analysis was performed for mature leaves and stems under stress, as there is no data displayed on Figure 5 for these plant tissues.
Response: Thank you for your valuable feedback on our manuscript. Because tissue-specific analysis showed that the gene was highly expressed in new leaves and roots, expression analysis under stress treatment was only performed on these two tissues.
- Lines 302-304. References are required.
Response: Thank you for your valuable feedback on our manuscript. We have added the corresponding references in lines 292 and 294.
- Lines 309-324. Improve the discussion of these data by discussing how the new data impacts the previously reported data, at least in other species.
Response: Thank you for your valuable feedback on our manuscript. We have improved the discussion section in the manuscript by adding discussions on how the new data impact the previously reported data in lines 298-305.
- Lines 342-344. References are required.
Response: Thank you for your valuable feedback on our manuscript. We have added the corresponding references in line 337.
- Lines 363-385. I am not sure if the model presented is fully supported. Upgrading the model using the main findings would be beneficial.
Response: Thank you for your valuable feedback on our manuscript. We have elaborated on the basis of the model in the discussion.
- Lines 397-400. Specifying the age of the seedling should be valuable.
Response: Thank you for your valuable feedback on our manuscript. We have specified the age of the seedling in lines 379-381.
- Lines 405-411. Specifying the number of days the plants were exposed to the different treatment is required. Also, recovery experiments are required. Sampling specification is required since Figure 4 displays localization data instead of tissue samples.
Response: Thank you for pointing out the issue in our manuscript. We have indicated the number of days of treatment in the figure, conducted recovery experiments, and provided supplementary explanations for the sample conditions in Figure 3.
- Lines 414-415. Specifying ing should be useful.
Response: Thanks for your suggestion. We have specified the sample in line 406.
- Section 4.5. All software use needs to be cited.
Response: Thanks for your suggestion. We have cited all the software used in the manuscript.
- Section 4.6. Specification of tissue sampling is required. Also, it is required to specify the amount of tissue used. Extended information on generating and processing RNA-qPCR data is required.
Response: Thank you for pointing out the issue in our manuscript. We have specified the specific methods for tissue sampling and clarified the amount of tissue used in Section 4.6.
- Lines 467-468. Specification of sequence information for primer is required.
Response: Thanks for your suggestion. The sequence information of the primers is shown in Table S1.
- Line 470-473. Verification that “Agrobacterium rhizogenes” is correct for the experiment mentioned.
Response: Thank you for your valuable feedback on our manuscript. We have verified that the mentioned “Agrobacterium rhizogenes” is correct.
- Lines 486-488. Specifying the soil material used is valuable.
Response: Thank you for pointing out the issue in our manuscript. We have specified the soil material in line 477-478.
- Section 4.10. The sampling of tissue requires specific specifications. Extended information on generating and processing RNA-qPCR data is required.
Response: Thank you for pointing out the issue in our manuscript. We have specified the detailed tissue sampling methods. We sincerely apologize for our inability to provide extended information on processing qPCR data.
- Consistency in the format of letters in the panels of the figure is recommended. See figure 3 compared to the other figures.
Response: Thank you for your valuable feedback on our manuscript. We have made the letter formats consistent across all panels in the figure.
- Consistency in the mentioned fusion of WRKY33 with GFP is required, for instance, 35S::MbWRKY33::GFP versus 35Spro::MbWRKY33::GFP.
Response: Thank you for pointing out the issue in our manuscript. We have standardized the description of the WRKY33-GFP fusion mentioned in the text, uniformly using 35S::MbWRKY33::GFP in line 168.
- It is suggested to have a clearly defined definition for sampling, data acquisition, and processing during RT-qPCR, and for each species. Determining the new and mature leaves.
Response: Thank you for your valuable feedback on our manuscript. We have supplemented clear definitions for sampling, data acquisition, and processing during the RT-qPCR procedure.
- The manuscript has typos. For instance, use of cursive genes (for instance line 96-98, 100-101). 113-117. Use cursive for species, for instance, line 435. Line 442.
Response: We are very sorry about our mistake. We have corrected the spelling mistakes in the manuscript.
We appreciate for Editors/Reviewers' warm work earnestly and hope that the correction will meet with approval.
Yours sincerely,
Deguo Han

Round 2
Reviewer 2 Report
Comments and Suggestions for Authors
The authors have addressed my previous concerns appropriately, and the manuscript has significantly improved in clarity and scientific rigor. I appreciate their efforts in revising the work. I have no further major concerns, and I believe the manuscript is now suitable for publication.
Author Response
尊敬的编辑和审稿人:
感谢您的来信和对我们修订后的手稿的评论。这些评论都很有价值,对修改和改进我们的论文非常有帮助,对我们的研究具有重要的指导意义。感谢您对我们修订稿件的认可。我们衷心感谢您对本文的批准和建设性的建议。
我们衷心感谢编辑/审稿人的热情工作,并希望更正能得到批准。
此致
韩德国
